# Responses of Foreign GA₃ Application on Seedling Growth of Castor Bean (*Ricinus communis* L.) under Salinity Stress Conditions

**Xiurong Jiao** [1,2,3], **Wenfang Zhi** [1,2,3], **Guijuan Liu** [1,2], **Guanglong Zhu** [1,2,4], **Gongneng Feng** [5], **Nimir Eltyb Ahmed Nimir** [4], **Irshad Ahmad** [1,2,3] **and Guisheng Zhou** [1,2,3,*]

1   Institutes of Agricultural Science and Technology Development of Yangzhou University, Yangzhou 225009, China; xiurongjiao2016@outlook.com (X.J.); wenfang.zhi@outlook.com (W.Z.); guijuan.liu@outlook.com (G.L.); g.zhu@yzu.edu.cn (G.Z.); Irshadgadoon48@gmail.com (I.A.)
2   Joint International Research Laboratory of Agriculture and Agri-Product Safety, the Ministry of Education of China, Yangzhou University, Yangzhou 225009, China
3   Jiangsu Provincial Key Laboratory of Crop Genetics and Physiology, Yangzhou University, Yangzhou 225009, China
4   Faculty of Agriculture, University of Khartoum, 11115 Khartoum, Sudan; nimir1000@gmail.com
5   Yancheng Institute of Technology, Yancheng 224002, China; ffyalce@ycit.cn
*   Correspondence: gszhou@yzu.edu.cn; Tel.: +86-133-3885-8866

**Abstract:** Castor bean (*Ricinus communis* L.), a promising bioenergy crop, is readily planted in marginal lands like saline soils. A controlled experiment was conducted to explore the possibility of using gibberellic acid (GA₃) as a promoter for caster bean grown under NaCl conditions and to try to determine the most appropriate concentration of GA₃ for seedling growth. The seeds of salt-tolerant cultivar Zibi 5 were firstly soaked with 0, 200, 250, and 300 μM GA₃ for 12 h and then cultured with 1/2 Hoagland solution containing 0, 50, and 100 mM NaCl in pots filled with sand. Plant height, stem diameter, leaf area, dry mater of each organ, activity of superoxide dismutase (SOD), peroxidase (POD) and catalase (CAT), soluble protein, and proline content in the leaves were examined. Plant height and stem diameter, SOD, and POD activity was significantly highest in the treatment of 250 μM GA₃ under salt concentration of 50 mM NaCl among all the testing days; protein content was highest when GA₃ concentration was 250 μM under 100 mM NaCl treatment. This indicated that caster bean seed soaking with 250 μM GA₃ could be the most suitable concentration for promoting seedling growth of caster bean, improving their stress resistance.

**Keywords:** castor bean; salinity stress; foreign GA₃ application; seedling growth and physiological parameters

## 1. Introduction

Castor bean (*Ricinus communis* L.) is an ideal oilseed crop because of its high seed oil content (more than 480 g kg⁻¹), unique fatty acid composition (900 g kg⁻¹ of ricinoleic acid) [1], and can be successfully mixed with petroleum diesel to reduce air pollution [2]. Due to these characteristics, caster bean has become one of the most promising candidates for biofuel production as well as a vital industrial raw material [3]. In addition, it can tolerate diverse weather conditions, especially, castor bean can be grown under drought and saline conditions [1,4,5]. All these features make it an attractive alternative biodiesel feedstock.

At present, China is rich in saline and alkaline land with 99 million hm² of saline soil and secondary saline soil and 2.2 million hm² of coastal mudflat resources [6], but most of them are undeveloped and

unutilized. One of the economical and efficient ways to improve and develop these saline soils is to screen salt tolerant oil crops [5].

Plants are sensitive to various abiotic stresses at the early stages of growth and development. Caster bean has been found to tolerate moderate salt stress during the later vegetative growth. By empirical observations, castor been seed germination seems not to be sensitive to salt stress [5]. However, it has been suggested that subsequent growth and development of seedlings are severely affected [7]. The threshold of Na salinity for castor emergence and growth is 7.1 dS m$^{-1}$. At this salinity level, 60% of the seedlings did not survive when subjected to the threshold salinity level [8].

Salinization has been an important environmental factor limiting crop growth and yield [9–11]. When plants are subjected to salt stress, reactive oxygen species (ROS) such as superoxide ($O_2{}^-$) and hydrogen peroxide ($H_2O_2$) produce as an initial response from the osmotic stress [12,13]. ROS has a detrimental effect on plants through oxidative damage to lipids, proteins, and nucleic acids [14]. However, plants have evolved a series of antioxidant systems that protect them from these potential cytotoxic effects. In which, one of the most effective of the ROS scavenging system is the antioxidant enzymes such as superoxide dismutase (SOD), peroxidase (POX), catalase (CAT), and ascorbate peroxidase (APX) [13,14]. Although there have been many reports on the changes of protective enzyme activities in plants under salt stress, few studies have been carried out on the castor bean seedling under salinity stress.

Proteins synthesis is affected by abiotic stresses, and salt-induced proteins have been identified [15]. Protein accumulated in the plants when grown under saline conditions. This might be because these accumulated proteins provide a storage form of nitrogen which could be reutilized when stress is over [16], and may also play a role in osmotic adjustment [17].

Proline is one of the main osmotic regulatory substances in plants under salt stress [18]. It has two main effects in anti-stress. Firstly, it is an osmotic regulatory substance that is used to maintain the osmotic balance between protoplasm and environment. It can form polymers with some intracellular compounds, similar to hydrophilic colloid, to prevent water loss. Secondly, it can reinforce the integrity of membrane structures.

Exogenous application of plant growth phytohormones has been regarded as an effective way to alleviate negative effects of salinity. Among these, gibberellins have been recorded to be a promoter for plant growth under salinity conditions [19–23] which can relieve seed dormancy, promote plant gene expression, increase the synthesis of hydrolase in plants, repair damaged cell membranes and improve seed vitality. Previous studies have shown that the use of exogenous gibberellic acid (GA$_3$) can promote seed germination, improve the salt tolerance of seeds, and alleviate the inhibition of salt on seedling growth [22,24–26]. It has also been concluded that usage of GA$_3$ reduces the harmful effects of salinity and increases resistance to salinity in the mustard plant [27].

The aim of this experiment was to evaluate the effects of salinity stress on caster bean at seedling development stage and try to find out the suitable concentration of GA$_3$ for alleviating the salt stress. We conducted a greenhouse experiment to study the different GA$_3$ concentration applications on plant height, stem diameter, leaf area, and dry matter of each part, then examined the superoxide dismutase (SOD), peroxidase (POD), and catalase (CAT) activity. The soluble protein and proline content were also tested with GA$_3$ amendment under the NaCl treatment at the initial stages of caster bean growth.

## 2. Materials and Methods

### 2.1. Plant Materials

Zibi 5 is a salt-tolerant castor bean cultivar, which was selected from Zibi 5, Zibi 7, and Zibi 8 based on a previous germination test in this study. Zibi 5 was selected because of its relative lower salt injury rate (which was calculated by germination rate, sprout length, and sprout thickness during the seed germination process) than Zibi 7 and Zibi 8 [28] under 100 mM NaCl. The best concentration and

time of GA$_3$ under different salt stress were also obtained from this previous study by soaking and culture under different salt concentrations [28].

## 2.2. Experimental Design

A pot experiment was conducted in a growth chamber at Joint International Research Laboratory of Agriculture and Agri-Product Safety of Ministry of Education of China, Yangzhou University, Jiangsu Province. The experiment was arranged in a factorial design with three salinity levels including 0, 50, and 100 mM NaCl; and four GA$_3$ concentrations including 0, 200, 250, and 300 μM. The castor bean cultivar used in this study was ZiBi 5, provided by Zibo Agricultural Research Institute in Shandong Province. There were three salinity levels and four GA$_3$ treatments repeated three times for a total of 36 pots for each time.

Homogeneous and healthy caster been seeds were selected and soaked with 0.1% HgCl$_2$ solution for 15 min for disinfection, and then rinsed with purified water three times. Then these castor seeds were soaked with different concentrations of GA$_3$ solution in darkroom for 12 h with a temperature of 25 °C, then the seeds were re-dried to near their original weight. Then, the seeds were buried in the wet sand containing 1/2 concentration of Hoagland solution for 48 h for pre-germination. Seeds with similar bud length were selected and planted in a plastic tray (50 cm long; 30 cm width; 5 cm high) with holes at the bottom. Each pot (5 cm in top diameter and 2.5 cm in bottom diameter) was filled with quartz sand. Three seeds were planted in each pot at 3 cm seeding depth.

The plastic tray was placed in a solution tank filled with 1/2 concentration of Hoagland solution with different NaCl concentrations. The plastic tray was then placed in a growth chamber. The temperature in the growth chamber was kept at 30/25 °C day/night (d/n) and 14/10 h d/n under a PAR of 500 W m$^{-2}$ with a relative humidity setting at 55%. The Hoagland solution was replaced once every five days.

## 2.3. Measurements

Plant materials were sampled on the 10, 20, and 30 days after planting. Ten plants were taken from each repetition and washed with distilled water. The length and width of cotyledon and true leaf, plant height, and stem diameter were measured. The stem diameter was measured 1 cm above the lateral root position. Roots were cleaned and washed with distilled water, and the plants were separated into roots, stems and leaves. Fresh weight was measured. The samples were divided into two parts. One part was put into an oven at 70 °C for 72 h until the plant material reached a constant weight for biomass determination. The other part of samples was stored in a −80 °C refrigerator after being frozen with liquid nitrogen for the determination of SOD, POD, CAT, soluble protein and proline content.

To measure enzymes activity, 0.5 g of fresh leaf of caster bean seedlings was cut into small pieces, quartz sand was added and mixed with 5 mL 50 mM/L phosphate buffer solution (pH = 7.8). The mixture was ground in ice condition, then centrifuged at 15,000 r/min for 20 min with the temperature of 4 °C. The supernatant was taken and kept at low temperature for further determination of SOD, POD, and CAT activity.

SOD activity was measured by monitoring the inhibition of the photochemical reduction of Nitrotetrazolium blue tetrazolium (NBT) spectrophotometrically at 560 nm [29]. The reaction mixture contained 50 mM of Na-phosphate buffer (pH 7.8), 750 μM of NBT, 130 μM of methionine, 100 μM of EDTA, 20 μM of riboflavin, ultrapure water and 0.1 mL of enzyme extract. The mixture was left for reaction under the fluorescent lamps (4000 Lux) for 30 min. Two controls (0.1 mL pH = 7.8 phosphate-buffered saline solution (PBS) was used to replace the enzyme solution) were placed in the dark. The absorbance was recorded at 560 nm. One unit of SOD enzyme activity was defined as the quantity of enzyme that reduced the absorbance reading of samples to 50% in comparison with tubes lacking enzymes.

The activity of POD was determined according to the method of Raza et al. [30]. A total of 0.1 mL of enzyme solution was added to the reaction solution (0.3% H$_2$O$_2$ 1.95 mL, 0.2% guaiacol 0.95 mL, pH = 7.0 PBS 1 mL). The enzyme solution was added to the reaction solution, with a pH = 7.8 PBS

as blank. The optical density (OD) increment of 0.01 per minute was defined as a vitality unit of POD activity.

CAT activity was determined according to the method of Raza et al. [30], estimating the absorbance decreased at 240 nm for 120 s, as a result of $H_2O_2$ consumption. Then, 0.1 mL of enzyme solution was added to the reaction mixture (0.3% $H_2O_2$ 1 mL, and 1.9 mL ultra-pure water). PBS (pH = 7.8) was added to the reaction mixture instead of the enzyme solution as blank for zero adjustment, and changes in absorbance of the reaction solution at 240 nm were read every 20 s. The reduction in OD of 0.01 per minute was defined as a unit of CAT activity.

The content of soluble proline was determined according to the method of Bates et al. [31], and the soluble protein was determined to the method of Coomassie Brilliant Blue G-250 Staining Method [32].

### 2.4. Statistical Analysis

Within a replicate, the average values across all the plants of each variable on a sampling date were calculated for each treatment. The data were then subjected to analysis of variance (ANOVA) using the statistical package DPS 7.05 for Windows (Beijing, China) [33] according to the two-factor randomized design. Treatment mean differences were separated by the least significant difference (LSD, $p$ = 0.05) test if the F-tests were significant.

## 3. Results

### 3.1. The GA₃ Effects on Caster Bean Seedling Growth under Different NaCl Treatments

The effects of NaCl, $GA_3$ and their interaction on plant height on 10 days (d), 20 d and 30 d were significant, as well as stem diameter and leaf area on 20 d and 30 d (Table S1).

Plant height under 100 mM NaCl was significantly lower than that under 0 and 50 mM NaCl, and plant height under 50 mM NaCl was slightly higher than that under 0 mM NaCl, but the difference was not significant (Table 1). The highest plant height of the seedling was under the 250 μM treatment of $GA_3$ under each salinity treatment during the sampling days (Table 1). On 10, 20, and 30 days after sowing, plant height was highest in the treatment with salt concentration of 50 mM NaCl and 250 μM GA3 with 25.33 and 27.74 cm on 20 d and 30 d, respectively, and lowest in the treatment with salt concentration of 100 mM NaCl and 300 μM $GA_3$ (14.72, 16.93, and 20.51, respectively) (Table 1). Thus, it seems that the promotion effects of $GA_3$ at 250 μM on plant height was much better than other $GA_3$ concentrations under all the salinity treatments.

**Table 1.** Effects of different levels of exogenous gibberellic acid ($GA_3$) on plant height of caster bean seedlings grown under three levels of NaCl treatment: 0 (control), 50, and 100 mM.

| | Salinity Level | GA₃ | 10 d | | 20 d | | 30 d | |
|---|---|---|---|---|---|---|---|---|
| | | 0 | 20.94 ± 0.75 | abcd | 21.31 ± 1.28 | de | 25.72 ± 1.21 | bc |
| | | 200 | 21.87 ± 0.32 | abc | 22.42 ± 0.14 | cde | 23.99 ± 0.29 | d |
| | 0 | 250 | 22.43 ± 0.52 | a | 24.67 ± 0.67 | ab | 27.32 ± 0.59 | ab |
| | | 300 | 20.59 ± 0.83 | cd | 22.8 ± 0.12 | cd | 23.59 ± 0.35 | d |
| | | 0 | 20.8 ± 0.78 | bcd | 22.92 ± 0.31 | c | 24.26 ± 0.3 | cd |
| Plant Height | | 200 | 21.55 ± 0.68 | abc | 22.98 ± 0.23 | c | 26.42 ± 1.24 | ab |
| (cm) | 50 | 250 | 22.25 ± 0.38 | ab | 25.33 ± 0.58 | a | 27.74 ± 0.48 | a |
| | | 300 | 21.87 ± 0.43 | abc | 23.67 ± 0.52 | bc | 26.59 ± 0.33 | ab |
| | | 0 | 17.39 ± 0.12 | e | 18.6 ± 0.36 | f | 22.8 ± 0.17 | de |
| | | 200 | 15.25 ± 0.37 | f | 16.55 ± 0.26 | g | 21.66 ± 0.53 | ef |
| | 100 | 250 | 19.89 ± 0.11 | d | 21.28 ± 0.15 | e | 23.08 ± 0.11 | de |
| | | 300 | 14.72 ± 0.66 | g | 16.93 ± 0.25 | g | 20.51 ± 0.66 | f |

Data are presented as mean ± standard deviation (SE) (*n* = 3). One-way ANOVA was conducted by Fisher's Least Significant Differences post hoc test. Values without letters in common are significantly different at the *p* = 0.05 level within each column for plant height. 10 d, 20 d, and 30 d represent 10, 20, and 30 days after seeding. LSD$_{0.05}$ = 1.605, 1.499, and 1.835 for 10 d, 20 d and 30 d, respectively.

The stem diameter of seedlings was increased at 200 μM GA$_3$ and then decreased with 300 μM GA$_3$ concentration. Caster been seedlings grown under 50 mM NaCl treatment with 250 μM GA$_3$ treatment showed the highest stem diameters (0.49, 0.52 and 0.57 cm on 10 d, 20 d and 30 d, respectively) than many other GA$_3$ concentration treatments (Table 2).

**Table 2.** Effects of different levels of exogenous GA$_3$ on stem diameter of caster bean seedlings grown under three levels of NaCl treatment: 0 (control), 50, and 100 mM.

|  | Salinity Level | GA$_3$ | 10 d | | 20 d | | 30 d | |
|---|---|---|---|---|---|---|---|---|
| Stem diameter (cm) | 0 | 0 | 0.40 ± 0.02 | de | 0.43 ± 0.00 | g | 0.48 ± 0.00 | d |
|  |  | 200 | 0.42 ± 0.02 | cde | 0.47 ± 0.01 | de | 0.48 ± 0.00 | d |
|  |  | 250 | 0.42 ± 0.01 | cde | 0.43 ± 0.00 | fg | 0.48 ± 0.01 | d |
|  |  | 300 | 0.39 ± 0.02 | e | 0.46 ± 0.01 | ef | 0.49 ± 0.01 | cd |
|  | 50 | 0 | 0.47 ± 0.02 | abc | 0.51 ± 0.01 | ab | 0.54 ± 0.01 | ab |
|  |  | 200 | 0.48 ± 0.01 | ab | 0.51 ± 0.01 | ab | 0.56 ± 0.02 | a |
|  |  | 250 | 0.49 ± 0.01 | a | 0.52 ± 0.00 | a | 0.57 ± 0.00 | a |
|  |  | 300 | 0.45 ± 0.01 | abc | 0.49 ± 0.01 | bcd | 0.52 ± 0.01 | bc |
|  | 100 | 0 | 0.40 ± 0.03 | de | 0.45 ± 0.01 | efg | 0.49 ± 0.02 | cd |
|  |  | 200 | 0.44 ± 0.02 | abcd | 0.48 ± 0.01 | cd | 0.51 ± 0.01 | bcd |
|  |  | 250 | 0.44 ± 0.01 | abcd | 0.51 ± 0.01 | ab | 0.57 ± 0.00 | a |
|  |  | 300 | 0.44 ± 0.01 | bcd | 0.50 ± 0.01 | bc | 0.54 ± 0.02 | ab |

Data are presented as mean ± SE (*n* = 3). One-way ANOVA was conducted by Fisher's Least Significant Differences post hoc test. Values without letters in common are significantly different at the *p* = 0.05 level within each column for stem diameter. 10 d, 20 d, and 30 d represent 10, 20, and 30 days after seeding. LSD$_{0.05}$ = 0.050, 0.025, and 0.034 for 10 d, 20 d and 30 d, respectively.

NaCl treatment significantly decreased leaf area (Table 3). The largest leaf area of caster bean seedling was found with the treatment of salinity concentration of 50 mM and GA$_3$ concentration of 250 μM on 20 d with a value of 43.92 cm$^2$ and with the treatment of salinity concentration of 0 mM and GA$_3$ concentration of 250 μM on 30 d with a value of 88.94 cm$^2$. The leaf area was the smallest when treated with salt concentration of 100 mM NaCl and GA$_3$ at 300 μM on 20 d and 30 d (Table 3).

**Table 3.** Effects of different levels of exogenous GA$_3$ on leaf area of caster bean seedlings grown under three levels of NaCl treatment: 0 (control), 50, and 100 mM.

|  | Salinity Level | GA$_3$ | 10 d | | 20 d | | 30 d | |
|---|---|---|---|---|---|---|---|---|
| Leaf area (cm$^2$) | 0 | 0 | 26.19 ± 2.83 | ab | 28.47 ± 1.94 | cd | 84.66 ± 2.89 | a |
|  |  | 200 | 26.94 ± 4.12 | ab | 32.18 ± 0.49 | bc | 48.93 ± 1.13 | c |
|  |  | 250 | 24.66 ± 2.41 | abc | 36.12 ± 0.82 | b | 88.94 ± 2.77 | a |
|  |  | 300 | 20.46 ± 2.22 | bcde | 32.59 ± 4.29 | bc | 63.41 ± 3.35 | b |
|  | 50 | 0 | 22.71 ± 2.34 | abcd | 34.8 ± 1.84 | b | 49.56 ± 2.94 | c |
|  |  | 200 | 28.57 ± 4.02 | a | 37.18 ± 2.92 | b | 53.66 ± 0.98 | c |
|  |  | 250 | 19.57 ± 0.54 | bcde | 43.92 ± 3.06 | a | 63.4 ± 0.91 | b |
|  |  | 300 | 22.19 ± 3.17 | abcd | 26.54 ± 2.97 | cd | 41.46 ± 2.61 | d |
|  | 100 | 0 | 17.11 ± 1.84 | cde | 24.13 ± 0.57 | d | 31.55 ± 1.68 | e |
|  |  | 200 | 14.33 ± 0.66 | e | 26.62 ± 0.67 | cd | 40.59 ± 0.6 | d |
|  |  | 250 | 19.55 ± 2.91 | bcde | 28.2 ± 0.88 | cd | 53.68 ± 1.75 | c |
|  |  | 300 | 15.84 ± 1.46 | de | 23.65 ± 0.38 | d | 23.65 ± 0.38 | f |

Data are presented as mean ± SE (*n* = 3). One-way ANOVA was conducted by Fisher's Least Significant Differences post hoc test. Values without letters in common are significantly different at the *p* = 0.05 level within each column for leaf area. 10 d, 20 d, and 30 d represent 10, 20, and 30 days after seeding. LSD$_{0.05}$ = 7.636, 6.242, and 6.083 for 10 d, 20 d, and 30 d, respectively.

The effects of salt treatment, GA$_3$ concentrations, and their interaction between them on the dry weight of each organ and the whole plant of castor bean seedlings on 10 d and 20 d was not significant.

There was a significant effect of NaCl on the dry matter of all the plant parts and whole plant on 30 d, while the effect of GA$_3$ on leaf and the whole plant was significant (Table S2). The interaction effect on the dry matter of root was significant on 30 d.

With the increase in salt concentration, the dry weight of leaves, stems, and the whole plant gradually decreased. The difference between 50 and 0 mM NaCl treatment was not significant, but significantly higher than that of 100 mM NaCl treatment. With the increase in GA$_3$ concentration, the dry weight of leaves and the whole plant increased first and then decreased, and the dry weight of leaves, stems, roots, and the whole plant treated with 250 μM GA$_3$ was the highest. On 30 d after sowing, seedlings treated with salt concentration of 0 mM NaCl and GA$_3$ concentration of 250 μM had the highest dry weight, while seedlings treated with salt concentration of 100 mM NaCl and GA$_3$ concentration of 300 μM had the lowest dry weight.

### 3.2. The GA3 Effects on SOD, POD and CAT under Different NaCl Treatments

Significant effects of NaCl and GA$_3$ and its interactions were found on SOD, POD, CAT, soluble protein, and proline content on all the sampling days (Table S3). GA$_3$ amendment with 250 μM significantly increased SOD activity, POD activity, and decreased CAT activity than that of the control treatment on 10 d, 20 d, and 30 d (Table 4).

With the increase in salt concentration, SOD activity increased first and then decreased. In general, SOD with 100 mM NaCl treatment was significantly lower than 0 mM NaCl treatment. On 20 d and 30 d after sowing, there was no significant difference in SOD activity between 0 and 50 mM NaCl treatments, but SOD activity of treatment with 100 mM NaCl was significantly lower than that of 0 and 50 mM NaCl treatments (Table 4). With the increase in GA3 concentration, SOD activity gradually increased and reached the maximum level at 250 μM GA$_3$ treatment, and then decreased at 300 μM under 0 and 50 mM NaCl. On 10 d, 20 d, and 30 d after sowing, SOD activity was highest in the treatment with salt concentration of 0 mM NaCl and GA$_3$ concentration of 250 μM of 1509, 573, and 519 U/g FW respectively, and lowest in the treatment with salt concentration of 100 mM NaCl and GA$_3$ concentration of 300 μM of 776, 405, and 274 U/g FW respectively.

With the increase in salt concentration, POD activity first increased at 50 mM NaCl and then decreased at 100 mM NaCl, and the amount of 100 mM NaCl was significantly lower than 0 mM NaCl (Table 4). The POD activity of GA$_3$ treatment was higher than that of the control. Under the conditions of 0 and 50 mM NaCl, POD activity gradually increased with the increase of GA$_3$ concentration to 250 μM, reaching the maximum, and then decreased slightly. POD activity was the highest in the treatment with a salt concentration of 0 mM and a GA$_3$ concentration of 250 μM (1995, 2648, and 3187 U/(g·min) on 10 d, 20 d, and 30 d, respectively), and the lowest in the treatment with a salt concentration of 100 mM and a GA$_3$ concentration of 200 μM (1055, 1107, and 1719 U/(g·min) on 10 d, 20 d, and 30 d, respectively) (Table 4).

No significant difference in CAT activity was found between the three NaCl concentrations, while 250 and 300 μM GA$_3$ treatment significantly lowered CAT activity than that of the control group under 0 mM and 50 mM NaCl conditions (Table 4). At 10, 20, and 30 d after sowing, CAT activity was highest in the treatment with 100 mM NaCl and 300 μM GA$_3$, and lowest in the treatment with 50 mM salt and 250 μM GA$_3$.

**Table 4.** Effects of different level of exogenous $GA_3$ on superoxide dismutase (SOD), peroxidase (POD), and catalase (CAT) of caster bean seedlings grown under three levels of NaCl treatment: 0 (control), 50, and 100 mM.

| | Salinity Level | GA3 | 10 d | | 20 d | | 30 d | |
|---|---|---|---|---|---|---|---|---|
| SOD (U/g FW) | 0 | 0 | 945 ± 22 | ef | 540 ± 23 | cd | 432 ± 19 | c |
| | | 200 | 1031 ± 21 | de | 564 ± 28 | bcd | 484 ± 15 | bc |
| | | 250 | 1509 ± 23 | a | 662 ± 23 | a | 596 ± 30 | a |
| | | 300 | 1151 ± 110 | c | 573 ± 14 | bcd | 519 ± 72 | abc |
| | 50 | 0 | 1024 ± 28 | de | 544 ± 25 | cd | 457 ± 27 | bc |
| | | 200 | 1360 ± 17 | b | 650 ± 38 | a | 582 ± 7 | a |
| | | 250 | 1340 ± 23 | b | 602 ± 34 | abc | 543 ± 11 | ab |
| | | 300 | 1356 ± 17 | b | 620 ± 20 | ab | 550 ± 13 | ab |
| | 100 | 0 | 836 ± 3 | fg | 506 ± 35 | d | 312 ± 45 | d |
| | | 200 | 800 ± 31 | g | 414 ± 30 | e | 285 ± 44 | d |
| | | 250 | 1079 ± 53 | cd | 534 ± 8 | cd | 509 ± 23 | abc |
| | | 300 | 776 ± 16 | g | 405 ± 10 | e | 274 ± 2 | d |
| POD (U/(g·min)) | 0 | 0 | 1360 ± 17 | g | 1411 ± 21 | ef | 2032 ± 26 | ef |
| | | 200 | 1515 ± 11 | cd | 1618 ± 112 | d | 2247 ± 206 | de |
| | | 250 | 1995 ± 54 | a | 2648 ± 80 | a | 3187 ± 24 | a |
| | | 300 | 1716 ± 26 | bc | 2350 ± 81 | b | 2659 ± 99 | b |
| | 50 | 0 | 1532 ± 8 | d | 2010 ± 16 | c | 2319 ± 40 | cd |
| | | 200 | 1750 ± 6 | b | 2426 ± 20 | b | 2648 ± 48 | b |
| | | 250 | 1968 ± 13 | a | 2629 ± 120 | a | 3045 ± 12 | a |
| | | 300 | 1631 ± 8 | c | 2153 ± 55 | c | 2538 ± 134 | bc |
| | 100 | 0 | 1242 ± 10 | g | 1349 ± 49 | ef | 1949 ± 95 | fg |
| | | 200 | 1055 ± 7 | h | 1107 ± 58 | g | 1719 ± 58 | g |
| | | 250 | 1451 ± 27 | f | 1514 ± 42 | de | 2119 ± 57 | def |
| | | 300 | 1219 ± 72 | g | 1305 ± 5 | f | 1923 ± 46 | fg |
| CAT (U/(g·min)) | 0 | 0 | 1026 ± 21 | bc | 404 ± 11 | cd | 384 ± 5 | d |
| | | 200 | 1041 ± 16 | ab | 382 ± 3 | d | 363 ± 12 | de |
| | | 250 | 640 ± 92 | e | 207 ± 17 | f | 183 ± 6 | i |
| | | 300 | 918 ± 54 | c | 225 ± 6 | f | 211 ± 6 | h |
| | 50 | 0 | 1024 ± 28 | bc | 355 ± 4 | de | 345 ± 8 | ef |
| | | 200 | 793 ± 40 | d | 322 ± 18 | e | 309 ± 2 | g |
| | | 250 | 638 ± 30 | e | 199 ± 13 | f | 173 ± 3 | i |
| | | 300 | 1024 ± 23 | bc | 345 ± 22 | e | 326 ± 12 | fg |
| | 100 | 0 | 1081 ± 6 | ab | 432 ± 7 | c | 413 ± 5 | c |
| | | 200 | 1083 ± 17 | ab | 484 ± 6 | b | 464 ± 10 | b |
| | | 250 | 1047 ± 25 | ab | 391 ± 5 | d | 374 ± 12 | d |
| | | 300 | 1154 ± 40 | a | 532 ± 19 | a | 520 ± 5 | a |

Data are presented as mean ± SE ($n$ = 3). One-way ANOVA was conducted by Fisher's Least Significant Differences post hoc test. Values without letters in common are significantly different at the $p$ = 0.05 level within each column for SOD, POD, and CAT activity, respectively. 10 d, 20 d, and 30 d represent 10, 20, and 30 days after seeding. $LSD_{0.05}$ (SOD) = 117.12, 75.34, and 93.02 for 10 d, 20 d, and 30 d, respectively; $LSD_{0.05}$ (POD) = 86.16, 191.12, and 257.07 for 10 d, 20 d, and 30 d, respectively; $LSD_{0.05}$ (CAT) = 114.45, 36.81, and 25.19 for 10 d, 20 d, and 30 d, respectively. 10 d, 20 d, and 30 d represent 10, 20, and 30 days after seeding.

### 3.3. The $GA_3$ Effects on Soluble Protein and Proline Content under Different NaCl Treatments

With the increase of salt concentration, the soluble protein and proline content in caster bean seedlings increased significantly on most of the sampling days. With the increase of $GA_3$ concentration, the soluble protein and proline content first increased and then decreased, with 250 μM of $GA_3$ treatment being the highest for all the NaCl concentrations. The soluble protein content was highest of 20.65 and 21.15 (μg/g FW) on 20 d and 30 d when the salt concentration was 100 mM NaCl and $GA_3$ concentration of 250 μM. The lowest soluble protein was found in the treatment with salt concentration of 0 mM NaCl and $GA_3$ of 0 μM (Table 5). For the proline content, treatment with $GA_3$ of 300 μM had

the significantly greater values of 41.56 and 27.92 (mg/g FW) under 100 mM NaCl treatment on 20 d and 30 d, respectively (Table 5).

**Table 5.** Effects of different levels of exogenous $GA_3$ on soluble protein and proline content of caster bean seedlings grown under three levels of NaCl treatment: 0 (control), 50, and 100 mM.

| | Salinity Level | GA3 | 10 d | | 20 d | | 30 d | |
|---|---|---|---|---|---|---|---|---|
| | | 0 | 24.54 ± 1.3 | f | 13.89 ± 0.59 | f | 15.49 ± 1.02 | def |
| | 0 | 200 | 25.65 ± 1.1 | f | 14.55 ± 0.84 | ef | 14.25 ± 0.83 | ef |
| | | 250 | 30.84 ± 0.43 | e | 16.14 ± 0.45 | def | 16.12 ± 0.78 | cde |
| | | 300 | 25.79 ± 0.96 | f | 16.28 ± 0.9 | cdef | 15.78 ± 0.61 | def |
| Soluble protein (µg/g FW) | | 0 | 35.23 ± 1.17 | cd | 14.89 ± 0.55 | def | 17.2 ± 0.61 | cd |
| | 50 | 200 | 42.74 ± 0.58 | b | 15.34 ± 1.48 | def | 18.34 ± 0.56 | bc |
| | | 250 | 47.64 ± 1.26 | a | 17.34 ± 1.06 | bcd | 20.23 ± 1.83 | ab |
| | | 300 | 34.88 ± 1.51 | cd | 18.74 ± 0.58 | abc | 18.58 ± 0.51 | bc |
| | | 0 | 32.53 ± 0.59 | de | 19.17 ± 0.22 | ab | 20.46 ± 0.61 | ab |
| | 100 | 200 | 30.18 ± 2.28 | e | 14.17 ± 1.23 | f | 13.49 ± 0.6 | f |
| | | 250 | 37.36 ± 0.67 | c | 20.65 ± 0.68 | a | 21.15 ± 0.81 | a |
| | | 300 | 20.12 ± 0.64 | g | 16.95 ± 1.07 | bcde | 13.87 ± 0.72 | ef |
| | | 0 | 15.08 ± 1.47 | d | 13.7 ± 1.9 | f | 12.82 ± 0.63 | f |
| | 0 | 200 | 16.75 ± 1.17 | d | 11.05 ± 0.3 | f | 13.24 ± 0.42 | ef |
| | | 250 | 17.45 ± 0.74 | d | 9.37 ± 0.25 | f | 13.03 ± 0.86 | f |
| | | 300 | 15.59 ± 1.91 | d | 12.56 ± 1.28 | f | 12.82 ± 0.63 | f |
| Proline content (mg/g FW) | | 0 | 23.61 ± 2.49 | c | 21.7 ± 2.04 | e | 15.7 ± 1.37 | def |
| | 50 | 200 | 35.18 ± 1.02 | b | 24.75 ± 1.26 | de | 17.63 ± 0.99 | cd |
| | | 250 | 37.05 ± 0.85 | b | 29.79 ± 1.53 | bc | 19.74 ± 0.33 | c |
| | | 300 | 31.67 ± 2.03 | b | 24.09 ± 2.68 | de | 16.26 ± 0.41 | de |
| | | 0 | 66.56 ± 2.82 | a | 26.32 ± 1.59 | cd | 20.61 ± 1.1 | c |
| | 100 | 200 | 34.71 ± 2.51 | b | 33.86 ± 0.28 | b | 24.26 ± 2.18 | b |
| | | 250 | 69.92 ± 3.29 | a | 23.9 ± 1.23 | de | 19.84 ± 1.35 | c |
| | | 300 | 31.48 ± 1.53 | b | 41.56 ± 2.11 | a | 27.92 ± 1.35 | a |

Data are presented as mean ± SE ($n = 3$). One-way ANOVA was conducted by Fisher's Least Significant Differences post hoc test. Values without letters in common are significantly different at the $p = 0.05$ level within each column for soluble protein and proline content, respectively. 10 d, 20 d, and 30 d represent 10, 20, and 30 days after seeding. $LSD_{0.05}$ (Soluble protein) = 3.368, 2.553, and 2.511 for 10 d, 20 d, and 30 d, respectively; $LSD_{0.05}$ (Proline content) = 5.788, 4.558, and 3.200 for 10 d, 20 d, and 30 d, respectively.

## 4. Discussion

Low level salinity (50 mM NaCl) had relatively small effects on plant height, leaf growth, and dry weight, and even had promoting effects, e.g., stem diameter of the caster bean seedlings was significantly increased than the control on 10 d, 20 d, and 30 d after sowing. However, high-level of salinity (100 mM NaCl) had significantly negative effects on castor seedling growth, which significantly reduced plant height, stem diameter, dry weight, and leaf area. Similar findings had been described that low concentration of salt (<0.3%) had little effect on the growth of caster beans and even had some promotional effect, while a high concentration of salt (>0.6%) significantly inhibited the growth of seedlings [8].

Caster bean seed soaked with 250 µM $GA_3$ significantly increased plant height, stem diameter, dry weight, and promoted the growth of leaves under all the NaCl treatments, while seed treated with other $GA_3$ concentration had little effect on seedling growth (Table 1). Previous study shows that gibberellic acid ($GA_3$) treatment has alleviated the negative effect of salinity in growth parameters in terms of leaf area, dry weight of grains, and photosynthetic pigments in wheat (*Triticum aestivum* L.) [34]. $GA_3$ enhanced plant growth and biomass yield of mustard (*Brassica juncea* L.) significantly [27], and significantly promoted plant length and plant fresh/dry biomass in soybean (*Glycine max* L.) while markedly hindered by NaCl induced salt stress [35]. Similar evidence was also reported that exogenous

application of gibberellic acid increased the total height and length of internodes in kenaf (*Hibiscus cannabinus* L.) plants at a low concentration (1.25 mL/L) [19]. It has been suggested that increased plant growth with $GA_3$ application can be related to the role of $GA_3$ in stimulating the cell elongation and division and internodal elongation [36,37], and the role in regulating the level of phytohormones [35].

In the present study, plant height and stem diameter were significant higher when castor plants were treated with $GA_3$ at a concentration of 250 μM than the other concentrations of $GA_3$ treatment. Seed treated with 300 μM $GA_3$ did not have a much stronger promoting effect on plant growth. This is similar to the results that increased level of gibberellin failed to increase the basal diameter, number of leaves, number of nodes, chlorophyll content, and biomass in all kenaf cultivars [19]. Another study also mentioned that the effect of 150 mg $L^{-1}$ $GA_3$ was much pronounced in improving grain yield when under salt stress compared with the other two concentrations (100 and 200 mg $L^{-1}$) [38]. This indicated that caster bean seed soaking with 250 μM $GA_3$ could be the most suitable concentration for promoting seedling growth of caster bean, improving their stress resistance and effectively alleviating the inhibition of salt stress.

Antioxidant enzymes play an important role in the scavenging of reactive oxygen species under salt stress. It has been demonstrated that activities of antioxidant defense enzymes changed in parallel with the increased $H_2O_2$ and salinity [39]. In the present study, compared with the 0 mM salinity level, the POD and SOD activity of castor bean seedlings at 50 mM salinity level was increased significantly at 10, 20, and 30 d after sowing (Table 4). This indicates that plant antioxidative mechanisms occurred in caster bean seedling for scavenging ROS under low-level salinity conditions. However, the POD and SOD activity of castor bean seedlings at the 100 mM salinity level were decreased significantly compared with that of 0 and 50 mM NaCl treatment (Table 4). It suggests that the high-level salinity might generate detrimental effects on the antioxidant system. It has been concluded that a severe salt stress often impairs the antioxidant system and causes oxidative stress [40].

At the 50 mM salinity level, compared with the 0 μM $GA_3$ treatment, caster bean seedlings with other level of $GA_3$ treatment significantly increased the activity of POD and SOD, and the POD and SOD activity of castor bean seedlings treated with 200 and 250 μM $GA_3$ reached the maximum (Table 4). This increased POD and SOD activity be considered as an improved protection which can be used as indicators of enhanced salt tolerance. Even though the activity of SOD and POD under 200 and 300 μM $GA_3$ treatment with 100 mM NaCl was significantly decreased, the enzyme activity under 250 μM $GA_3$ treatment remained at a high level (Table 4). Thus, it is concluded that $GA_3$ treatment with an appropriate concentration can enhance the protective enzyme activity in castor bean seedling leaves. Therefore, it is concluded in the present study that $GA_3$ treatment increased the salt tolerance of castor bean seedlings, partly via improving the antioxidant system.

This study also found that with the increase of salt concentration, the activity of CAT decreased after the increasing trend. The change of CAT activity was inversely related with the change of POD and SOD activity (Table 4). A similar result has been found that CAT activity decreased 61.27% when treated with a 7 dS $m^{-1}$ salinity level compared with the control in wheat [41]. This study suggested that SOD and POD probably played a key role in scavenging $O_2$ and $H_2O_2$ at low level salinity, and CAT may play an important role in scavenging high level of $H_2O_2$. In order to improve the salt tolerance of plants, it is crucial to maintain a good balance and coordinated expression of several antioxidant enzymes in plants. It might be more reasonable to take the good balance and coordinated expression among SOD, POD, and CAT as the indicator to judge the salt tolerance of plants.

Proline is an important osmolyte and an indicator of stress, as it can reestablish an osmotic balance under stress conditions including salinity [18]. Soluble protein is closely related to plant stress resistance. It has been found that protein synthesis can be triggered in plants under salt stress [42]. Therefore, the increase of salt stress proteins can be considered as a mechanism of plant self-protection and salt resistance. These proteins may be synthesized de novo in response to salt stress or may be present constitutively at a low concentration and increase when plants are exposed to salt stress [43].

In the present study, with the increased salinity, the content of proline and soluble protein increased at 10, 20, and 30 d after seeding (Table 5). It has been mentioned that salt stress had no significant effect on the concentration of soluble proteins at 50 mM NaCl; however, 100 mM NaCl treatment induced a significant increase in soluble proteins in the leaves of paper mulberry (*Broussonetia papyrifera* L.) [39]. Another study showed that salinity stress induced a significant increase in the soluble protein and proline content in root and leaf of two wheat cultivars (*Triticum aestivum* L.) [34]. Proline content can be used as indicator for evaluating the salt tolerance of *Lonicera japonica* Thunb [44]. After short treatment, the content of proline in the leaves of treelike *Lonicera japonica* Thunb. increased dramatically with the increased concentration of NaCl. Proline content was significantly increased in soybean under 50 to 200 mM NaCl [45]. Thus, it further confirms that caster bean seedlings are sensitive to NaCl stress due to the significantly increased soluble protein and proline content under both 50 and 100 mM NaCl. It has been suggested that proline showed stronger osmotic regulation ability under salinity stress [44]. Our results also confirmed this.

It has been suggested that a high concentration of proline acts as a soluble solute for inter cellular osmotic adjustment [46,47]. This study found that the change of soluble protein content and proline content of the changes is similar under salt stress. At the 50 mM salinity level, with increased $GA_3$ level, the content of soluble protein increased first then decreased; proline content increased with the increase of $GA_3$ concentration under low-salt conditions. The soluble protein content was significantly highest when $GA_3$ concentration was 250 μM under 100 mM NaCl treatment (Table 5). For the proline content, treatment with $GA_3$ of 300 μM had the significantly largest values under 100 mM NaCl treatment on 20 d and 30 d, indicating that protein and proline could accumulate under salt stress, and $GA_3$ treatment was helpful for its accumulation. It has been concluded that the increase in SOD and POD activity and proline content decreased the adverse effects of salinity stress on studied cultivars of wheat [41].

Glycine betaine was shown to improve salt tolerance in *Zea mays* L. cv. Giza by protecting chloroplast membrane, preserving photosynthetic pigments, enhancing osmotic adjustment, and turgor maintenance by accumulating organic osmolytes (total amino nitrogen, proline, total soluble protein, and total soluble sugars) [47]. From our results, the application of $GA_3$ under NaCl treatment might improve salt tolerance of caster bean through accumulating soluble protein and proline in the seedling. Similar studies have found that $GA_3$ treatments activated, in most cases, the production of soluble protein in the different organs of the two wheat cultivars as compared with the corresponding salinized plants [34]. An interactive effect of Ascorbic acid (AsA) with $GA_3$ enhanced protein content and the activity of the antioxidant enzyme in common bean plants under salt stress [48]. It has been concluded that $GA_3$ treatment increased adaptation of two wheat cultivars to salinity by osmoregulation using the organic solutes (saccharides and proteins) [34]. Thus, it was concluded that $GA_3$ treatment has alleviated the drastic effect of salinity in caster bean seedlings via protein and proline accumulation in the leaves.

The increase in soluble protein has a certain role in promoting caster bean seedlings to adapt to the high salt environment. Proline is also a ROS scavenger and a source of nitrogen and carbon required for stress recovery [49]. Furthermore, the interaction between proline and protein can increase the solubility of protein, reduce the precipitation of soluble protein, and enhance the hydration of protein. This has illustrated that the accumulation of proline could help to maintain the integrity and the function of the barley vacuole membrane, thus leading to enhance the ability to resist salt damage [50].

## 5. Conclusions

An appropriate amount of $GA_3$ has been reported in many crops to mitigate the harmful effect of abiotic stress. However, the exact concentration of $GA_3$ to make this alleviation for caster beans has not been studied previously. Our study examined many $GA_3$ concentrations on ameliorating gradual levels of salinity. In our study, plant height and stem diameter, SOD activity and POD activity were significantly highest in the treatment of 250 μM $GA_3$ under salt concentration of 50 mM NaCl among

all the treatments. The soluble protein content was highest when GA$_3$ concentration was 250 μM under 100 mM NaCl treatment. GA$_3$ alleviated salinity stress on castor seedlings by regulating the activity of antioxidase and accumulating the content of proline and soluble protein. Based on these results, it is concluded that caster bean seed soaking with 250 μM GA$_3$ could be the most suitable concentration for promoting seedling growth of caster beans, improving their stress resistance and effectively alleviating the inhibition of salt stress.

**Supplementary Materials:** The following are available online at http://www.mdpi.com/2073-4395/9/6/274/s1, Table S1: F values from ANOVA for plant height, stem diameter, leaf area related to NaCl, GA$_3$ and their interactions of castor bean seedlings, Table S2: F values from ANOVA for dry matter related to NaCl, GA$_3$ and their interactions of castor bean seedlings, Table S3: F values from ANOVA for SOD, POD and CAT activity, soluble protein and proline content related to NaCl, GA$_3$ and their interactions of castor bean seedlings.

**Author Contributions:** G.Z. (Guisheng Zhou) applied for the funding, administrated the project and designed the study. W.Z and G.L. conducted the experiments. X.J. prepared the data, undertook the analysis and wrote the manuscript. G.Z. (Guanglong Zhu) and G.F., N.E.A.N. and I.A. provided direction and supervised the work. All authors contributed to editing the manuscript.

**Funding:** This research was funded, in part, by Jiangsu Provincial Key R & D Program (BE2016345), Jiangsu Provincial Agricultural Innovation Fund (CX16(1005)), and Jiangsu Provincial Modern Agricultural Industrialization Development Program (2019).

**Conflicts of Interest:** The authors declare no conflict of interest.

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
