# Peer review of "Responses of Foreign GA3 Application on Seedling Growth of Castor Bean (Ricinus communis L.) under Salinity Stress Conditions"

_agronomy, doi:10.3390/agronomy9060274_

Reviewer 1 Report

The manuscript presents some inaccuracies and above all it does not appear very clear in the exposition of the results and in the organization and caption of the tables. In particular, tables 1, 2 and 6 named “Analysis of variance”, in fact, report only the value of F and are therefore not complete. Another critical aspect is related to the succession of the tables themselves that do not fully respond to the order in which they are cited in the text (table 2 should be postponed as table 5).

This imprecision in the exposure of the tables (it is not clear why in some tables the first line is partially underlined and in bold) has repercussions in the same understanding of the results, which at the moment does not appear suitable.

Finally, in the discussion reference is made to glycine betanin which was not used in the test and is not well connected to the analyzed experimental factor (GA3).

Among the inaccuracies in the text remember:

lines 17 and 34: Ricinus communis and not Ricinus Communis

line 88: it is better to use the term cultivar instead of variety and the correct denomination of the cultivars themselves requires that they be written in single quotation marks (eg ‘Zibi 5’)

Finally, there are numerous inaccuracies in the bibliography.

Author Response

Comments and Suggestions for Authors

The manuscript presents some inaccuracies and above all it does not appear very clear in the exposition of the results and in the organization and caption of the tables. In particular, tables 1, 2 and 6 named “Analysis of variance”, in fact, report only the value of F and are therefore not complete. Another critical aspect is related to the succession of the tables themselves that do not fully respond to the order in which they are cited in the text (table 2 should be postponed as table 5).

This imprecision in the exposure of the tables (it is not clear why in some tables the first line is partially underlined and in bold) has repercussions in the same understanding of the results, which at the moment does not appear suitable.

Answer: This has been modified and updated. We report only F value just to showed the significant differences within each factor, and table will become so big if reported the others item of ANOVA table.

Finally, in the discussion reference is made to glycine betanin which was not used in the test and is not well connected to the analyzed experimental factor (GA3).

Answer: Yes, Glycine betanin is not exactly the same as GA3, but we suggested that GA3 might reduce salt stress just like Glycine betaine, by enhancing osmotic adjustment and accumulating soluble protein and proline.

Among the inaccuracies in the text remember:

lines 17 and 34: Ricinus communis and not Ricinus Communis

Answer: Done.

line 88: it is better to use the term cultivar instead of variety and the correct denomination of the cultivars themselves requires that they be written in single quotation marks (eg ‘Zibi 5’)

Answer: Done. I have corrected them in the text.

Finally, there are numerous inaccuracies in the bibliography.

Answer: Done. We have checked the reference and updated them.

Reviewer 2 Report

Overall, I thought this manuscript addressed an interesting crop. I do suggest the manuscript be reviewed for grammar and diction. My comments below are meant to help provide clarity for the readers (as well as myself).

Introduction, Material and Methods

Line 88: Zibi5 was selected from Zibi 7, Zibi 8 and Zibi 5? Was Zibi 5 the progeny of a cross between Zibi 7 and 8? Or are the authors using this sentence as a rationale for why they used Zibi 5 for this experiment instead of Zibi 7 and 8? The authors may wish to rephrase this sentence to avoid confusing the audience.

Line 88-91: Which concentrations and time points were tested? Are these tests published? If so, cite. The authors may wish to consider including these preliminary experiments as supplementary material.

Line 97: Separate “including” and “0” by a space.

Line 101: Did the authors use mercury (I) chloride (Hg2Cl2) or mercury (II) chloride (HgCl2)? From what I understand, HgCl is diamagnetic. It cannot have unpaired electrons and exists as Hg2Cl2.

Line 105: sand not sands Please check the entire manuscript for similar inconsistencies.

Line 105: Separate “48” and “h” with a space. Please check the document for similar inconsistencies.

Line 109: The authors refer to a plastic disc (line 109) then a plastic tray (line 110). Is this the same object?

Line 112: What was the photosynthetic photon flux density at canopy level (µmol m-2 s-1)?

Line 113: days or d? Please review the manuscript for similar inconsistencies.

Line 144: “1.9 mL” not “1.9ml”

Line 155: The authors tested for cadmium (Cd) concentrations?

Results

I am not convinced that you need to include your ANOVA tables in the body of this manuscript (Tables 1, 2, & 6). Consider moving to the Supplementary Materials. For the data tables (3, 4 & 5), please include your LSD values. In the notes section below each table please include the type of test. For example, a two way ANOVA with a Fisher’s Least Significant Differences post hoc test. In tables 3, 4 and 5 – why are the 10 d 0 GA3 and part of the 20 d 0 GA3 treatments underlined and bolded?

Based on the tables, the readership can determine if one treatment is significantly higher or lower based on the provided tables. It can be helpful when reporting results to make comparisons between treatments of interest. For example, “treatment A was 3-fold higher than treatment B”. Alternatively, “treatment C resulted in 30 % more biomass as compared with plants sprayed with treatment F.”

I found the lettering scheme in Figures 1 and 2 difficult to follow. Why are Figures 1 and 2 not presented in tables? They have similar treatment groups as the data presented in Tables 3, 4 & 5. Or the authors may wish to present the data as dose response curves. Either way, I would recommend not using bar charts (as presented).

Lines 159-onwards: Would it be possible to refer to the time points as 10 d, 20 d or 30 d instead of d10, d20, and d30? This would be consistent with the descriptions of the GA3 and NaCl treatments.

Line 169: The “3” in GA3 should be in lower case

Discussion and conclusions

Line 285: why is gibberellic acid defined again?

Did the authors consider running Pearson correlations to determine if there was a relationship between the GA treatments and the alleviation of salt toxicity in the measured physiological or biochemical parameters? This would help to tie together some of the points mentioned in the discussion.

Line 386: exact not exactly

Author Response

Comments and Suggestions for Authors

Overall, I thought this manuscript addressed an interesting crop. I do suggest the manuscript be reviewed for grammar and diction. My comments below are meant to help provide clarity for the readers (as well as myself).

Introduction, Material and Methods

Line 88: Zibi5 was selected from Zibi 7, Zibi 8 and Zibi 5? Was Zibi 5 the progeny of a cross between Zibi 7 and 8? Or are the authors using this sentence as a rationale for why they used Zibi 5 for this experiment instead of Zibi 7 and 8? The authors may wish to rephrase this sentence to avoid confusing the audience.

Answer: Zibi 5 was selected from a preliminary experiment, it is not a cross between Zibi 7 and Zibi 8.  Zibi 5 was selected because of its relative lower salt injury rate (which was calculated by germination rate, sprout length and sprout thickness during the seed germination process) than Zibi 7 and Zibi 8 (data not shown) under 100 mM NaCl.

Line 88-91: Which concentrations and time points were tested? Are these tests published? If so, cite. The authors may wish to consider including these preliminary experiments as supplementary material.

Answer: The results will be published later, so I did not mention too much details here. But I have included the essential things here for reader’s understanding.

Line 97: Separate “including” and “0” by a space.

Answer: Done.

Line 101: Did the authors use mercury (I) chloride (Hg2Cl2) or mercury (II) chloride (HgCl2)? From what I understand, HgCl is diamagnetic. It cannot have unpaired electrons and exists as Hg2Cl2.

Answer: We did use HgCl2. I have corrected it in the main context. Thank you for your observation,

Line 105: sand not sands Please check the entire manuscript for similar inconsistencies.

Answer: Done. I have corrected all of them throughout the paper.

Line 105: Separate “48” and “h” with a space. Please check the document for similar inconsistencies.

Answer: Done. I have corrected all of them throughout the paper.

Line 109: The authors refer to a plastic disc (line 109) then a plastic tray (line 110). Is this the same object?

Answer: Yes, they are the same thing. I should use plastic tray for consistency. This has been corrected.

Line 112: What was the photosynthetic photon flux density at canopy level (µmol m-2 s-1)?

Answer: We did not test that exactly.

Line 113: days or d? Please review the manuscript for similar inconsistencies.

Answer: Done.

Line 144: “1.9 mL” not “1.9ml”

Answer: Done.

Line 155: The authors tested for cadmium (Cd) concentrations?

Answer: No. This is a mistake.

Results

I am not convinced that you need to include your ANOVA tables in the body of this manuscript (Tables 1, 2, & 6). Consider moving to the Supplementary Materials. For the data tables (3, 4 & 5), please include your LSD values. In the notes section below each table please include the type of test. For example, a two way ANOVA with a Fisher’s Least Significant Differences post hoc test. In tables 3, 4 and 5 – why are the 10 d 0 GA3 and part of the 20 d 0 GA3 treatments underlined and bolded?

Answer: We have moved the Table 1,2 and 6 to the Supplementary part. And they have been marked as Table S1, Table S2 and Table S3; We have included all the LSD values; Now we deleted the bold from (10 d 0 GA3 and part of the 20 d 0 GA3).

Based on the tables, the readership can determine if one treatment is significantly higher or lower based on the provided tables. It can be helpful when reporting results to make comparisons between treatments of interest. For example, “treatment A was 3-fold higher than treatment B”. Alternatively, “treatment C resulted in 30 % more biomass as compared with plants sprayed with treatment F.”

I found the lettering scheme in Figures 1 and 2 difficult to follow. Why are Figures 1 and 2 not presented in tables? They have similar treatment groups as the data presented in Tables 3, 4 & 5. Or the authors may wish to present the data as dose response curves. Either way, I would recommend not using bar charts (as presented).

Answer: Good suggestion we change the figure 1 and 2 to table.

Lines 159-onwards: Would it be possible to refer to the time points as 10 d, 20 d or 30 d instead of d10, d20, and d30? This would be consistent with the descriptions of the GA3 and NaCl treatments.

Answer: Yes, it would be better to use 10 d, 20 d and 30 d. I have updated that.

Line 169: The “3” in GA3 should be in lower case

Answer: Done.

Discussion and conclusions

Line 285: why is gibberellic acid defined again?

Answer: We did not get the points of this question.

Did the authors consider running Pearson correlations to determine if there was a relationship between the GA treatments and the alleviation of salt toxicity in the measured physiological or biochemical parameters? This would help to tie together some of the points mentioned in the discussion.

Answer: good question, we mention and hypotheses  that GA3 mitigated the negative impact of some parameters like leaf area, grain yield and photosynthesis parameters because all these parameters decline at level of salt concentration and when treated the crop by GA3 at the same salt level concentration, the mention parameters will be increased.

Line 386: exact not exactly

Answer: Done.

Reviewer 3 Report

I have carefully read and reviewed this manuscript. This manuscript is very interesting and presents important data concerning the responses of the gibberellin application on castor bean seedlings under salinity stress to improve the plant growth.

The approach, results and conclusions are intelligible from the abstract.

The title is informative, clear and it is in good correlation with the article content.

This paper is, in general, clearly written, using correct grammar and syntax.

However, the critical point of the manuscript is that the authors only assessed a salt-tolerant genotype without considering a non salt-tolerant genotype.

In my opinion, manuscript need a major revision. The addition of another experiment considering also a non salt-tolerant genotype is required or point out both in the title and abstract that the work was performed using a salt-tolerant genotype and finally reporting in the conclusion that further investigation are needed testing a non salt-tolerant genotype.

Line 17. Ricinus Communis L.

When the acronyms appear for the first time, these should be reported out in full

Line 26. Why all treatment?

Line 63. This sentence is not clear.

Line 88. “a salt-tolerant”. Why the author assessed only a salt-tolerant genotype?

Line 88-89. This sentence is not clear.

The authors should report the threshold values of pH and salinity for the assessed genotype.

The authors should report the pH and salinity (EC) of each assessed treatment.

Line 97. Add a space between including and 0.

Line 152. Did the authors verify the assumptions of the ANOVA.

When the name of each crop appears for the first time, these should be also reported using the scientific name.

Line 300. “under salt stress”. OK, but which level?

Line 345. Which cultivar of wheat?

Lines 386-387. This sentence is not clear.

Author Response

Comments and Suggestions for Authors

I have carefully read and reviewed this manuscript. This manuscript is very interesting and presents important data concerning the responses of the gibberellin application on castor bean seedlings under salinity stress to improve the plant growth.

The approach, results and conclusions are intelligible from the abstract.

The title is informative, clear and it is in good correlation with the article content.

This paper is, in general, clearly written, using correct grammar and syntax.

However, the critical point of the manuscript is that the authors only assessed a salt-tolerant genotype without considering a non salt-tolerant genotype.

Answer: we would like to use the salt-tolerant genotype to find the suitable GA3 concentration when grown on salinity land, this experiment was carried out for providing field production guides. Yes, it would be better to include a non-salt-tolerant genotype just for comparison or examine the effect on that. We can consider that in the subsequent experiments.

In my opinion, manuscript need a major revision. The addition of another experiment considering also a non salt-tolerant genotype is required or point out both in the title and abstract that the work was performed using a salt-tolerant genotype and finally reporting in the conclusion that further investigation are needed testing a non salt-tolerant genotype.

Answer: Yes, it would be better to include a non-salt-tolerant genotype just for comparison or examine the effect on that. We can consider that in the subsequent experiments.

Line 17. Ricinus Communis L.

When the acronyms appear for the first time, these should be reported out in full

Answer: Yes.

Line 26. Why all treatment?

Answer: it should be among all the testing days: 10 d, 20 d and 30 d.

Line 63. This sentence is not clear.

Answer: We have modified this sentence.

Line 88. “a salt-tolerant”. Why the author assessed only a salt-tolerant genotype?

Answer: it was selected based on a previous seed germination test.

Line 88-89. This sentence is not clear.

The authors should report the threshold values of pH and salinity for the assessed genotype.

The authors should report the pH and salinity (EC) of each assessed treatment.

Answer: Unfortunately, in this study, we did not test the threshold values of pH and salinity for Zibi 5.

Line 97. Add a space between including and 0.

Answer: Done.

Line 152. Did the authors verify the assumptions of the ANOVA.

Answer: yes

When the name of each crop appears for the first time, these should be also reported using the scientific name.

Answer:  We revise it

Line 300. “under salt stress”. OK, but which level?

Answer: This has been modified.

Line 345. Which cultivar of wheat?

Answer: This has been updated.

Lines 386-387. This sentence is not clear.

Answer: This has been modified.

Round  2

Reviewer 1 Report

The authors corrected the manuscript according to the suggestions given. Moving some tables to supplementary materials and transforming some figures into tables makes it easier to understand the experiment. The only thing I would suggest is the modification of the table captions, now present in the supplementary materials; the tables not indicate the analysis of variance but the “F values from ANOVA for” (different parameters …).

Author Response

Reviewer 1: 

The authors corrected the manuscript according to the suggestions given. Moving some tables to supplementary materials and transforming some figures into tables makes it easier to understand the experiment. The only thing I would suggest is the modification of the table captions, now present in the supplementary materials; the tables not indicate the analysis of variance but the “F values from ANOVA for” (different parameters …).

Response 1: Good suggestions, we have changed the table captions to F values from ANOVA for plant height, stem diameter……(other parameters).

Reviewer 3 Report

Throughout the review, the authors have made significant improvements to the manuscript.

Author Response

Reviewer 3:

Throughout the review, the authors have made significant improvements to the manuscript.

Response : Thank you for your approve.